# Tabero: Learning Gentle Manipulation with Closed-Loop Force Feedback from Vision, Touch, and Language

**Qiwei Wu** [1 2]  **Rui Zhang** [1]  **Xin Xiang** [2]  **Tao Li** [1]  **Weihua Zhang** [1]  **Junjie Lai** [1]  **Renjing Xu** [2]

## Abstract

Tactile sensing is essential for robots to achieve human-like gentle manipulation. However, existing Vision-Language-Action (VLA) models struggle to exploit tactile feedback for gentle manipulation due to scarce aligned vision-tactile-language data and the lack of effective closed-loop force feedback mechanisms. To address these challenges, we introduce Tabero, a benchmark and model suite for gentle, language-conditioned robotic manipulation that demands fine-grained contact force perception. First, the Tabero benchmark addresses the scarcity of tactile data by presenting a data-efficient pipeline that repurposes open-source robot manipulation trajectories to generate diverse vision-tactile-language tasks, and establishes a multidimensional evaluation protocol that measures task success alongside physical interaction quality. Second, we propose Tabero-VTLA, an architecture with a decoupled force-position command interface; the resulting force-position commands are executed by a fixed hybrid controller to enable real-time, force-aware manipulation. Evaluated on Tabero, our model maintains high task success while reducing average grip force by over 70% under gentle instructions, demonstrating its ability to modulate interaction forces based on multimodal experience.

## 1. Introduction

Physical AI is emerging as a pivotal enabler for robots to operate effectively in the real physical world. For robots to exhibit genuine intelligence in unstructured environments, they must not only perceive their surroundings visually but also comprehend physical laws through direct contact. In humans, touch serves as the most fundamental modality for interacting with objects and is essential for developing physical intuition. While recent advances in vision–language–action (VLA) foundation models have shown remarkable progress (Kim et al., 2025a; Black et al., 2025b;a; NVIDIA et al., 2025; Liu et al., 2025), these models predominantly rely on internet-scale image–text–video data or robot datasets consisting of image–action pairs collected via specialized hardware. Crucially, they lack tactile modality altogether, limiting their ability to perform force-sensitive tasks such as gentle object handling. Although a few studies (Zhao et al., 2025; Wu et al., 2025; Cheng et al., 2026) have gathered real tactile data using custom-built hardware, the high cost, maintenance complexity, and low data collection efficiency of such systems make it extremely challenging to construct large-scale tactile datasets. Simulation offers a scalable alternative, yet existing pipelines focus on visual diversity and lack efficient mechanisms to generate and integrate high-fidelity tactile signals.

Building upon VLA models, vision–tactile–language–action (VTLA) models extend perception to include the tactile modality, thereby endowing foundation models with the capacity to interact physically with the world. Training such models, however, faces two challenges. First, VTLA models still require vast amounts of vision–tactile manipulation data. Second, there is no standardized benchmark to evaluate model performance at the level of physical interaction. Existing evaluation protocols based on task success rates focus exclusively on outcomes and overlook critical aspects of the interaction process, such as whether objects are damaged or excessive forces are applied during manipulation.

To enable language-conditioned gentle manipulation, we introduce Tabero (Fig. 1), a benchmark and model suite that tackles data scarcity and the absence of force-aware control in existing VLA systems. Tabero repurposes open-source robot trajectories via tactile simulation to generate diverse vision-tactile-language datasets and introduces a multidimensional evaluation protocol measuring both task success and interaction gentleness. Built on this framework, our Tabero-VTLA model suite integrates tactile observations into a VTLA architecture, outputting coordinated force-

---

[1]Nvidia, Beijing, China [2]Hong Kong University of Science and Technology (Guangzhou), Guangzhou, China. Correspondence to: Junjie Lai <julienl@nvidia.com>, Renjing Xu <renjingxu@hkust-gz.edu.cn>.

*Proceedings of the 43rd International Conference on Machine Learning*, Seoul, South Korea. PMLR 306, 2026. Copyright 2026 by the author(s).

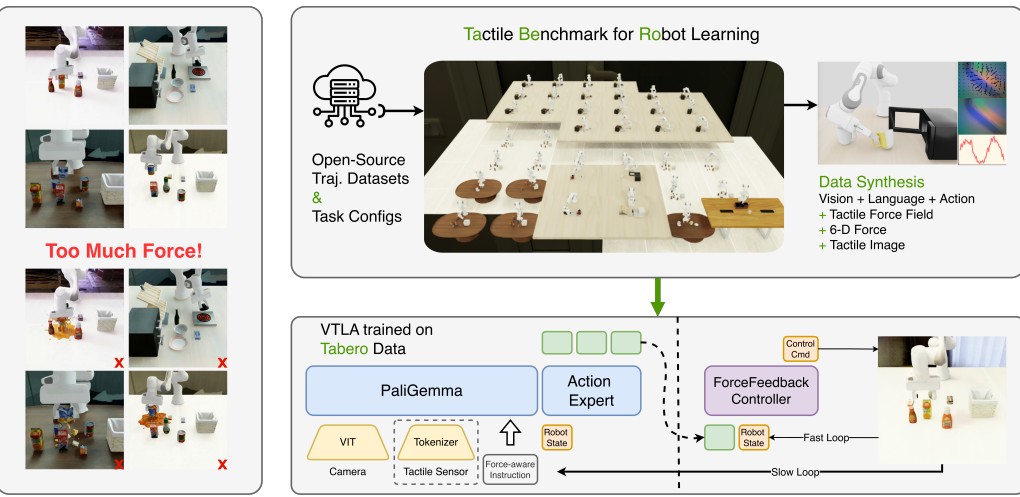

*Figure 1.* **Overview of the proposed framework. Motivation:** Current vision–language–action (VLA) systems and robotic arm–gripper setups based on synthetic data lack force feedback mechanisms, causing learned policies to frequently damage objects during manipulation. **Tabero:** We present a high-fidelity multimodal simulation platform integrating Isaac Lab with advanced tactile simulation. Our pipeline enables the re-collection of open-source datasets to generate synchronized streams of multi-view vision, tactile images, force fields, and proprioception. **Tabero-VTLA:** Leveraging the Tabero dataset, we propose a VTLA system featuring a decoupled force–position controller and introduce a multidimensional evaluation protocol to comprehensively assess the quality of physical interaction.

position commands that a compliant low-level controller executes for gentler manipulation. Experiments show it significantly reduces interaction forces while maintaining high task success.

In summary, our work makes the following contributions:
**The Tabero benchmark**, which enables scalable vision-tactile-language data generation by replaying open-source trajectories in a high-fidelity tactile simulator and establishes the first standardized protocol for quantifying gentleness in language-conditioned manipulation.
**Tabero-VTLA**, a suite of force-aware VLA models that introduce a decoupled force-position command interface to enable gentler interactions through substantially reduced contact forces while preserving high task success.

**Conflict of Interest Disclosure.** No conflict of interest exists.

## 2. Related works

**Simulation Platforms and Synthetic Data.** Simulated environments for robotic manipulation have seen significant progress in recent years. The LIBERO (Liu et al., 2023) benchmark systematically introduced a lifelong learning evaluation suite for robot manipulation tasks, built upon the RoboSuite framework to provide a comprehensive assessment protocol. RoboCasa (Nasiriany et al., 2024) extends RoboSuite (Zhu et al., 2020) using the MuJoCo physics engine and delivers extensive human demonstra-

tion data along with procedural generation methods in kitchen scenarios. RoboTwin (Mu et al., 2025; Chen et al., 2025) offers a data generation pipeline tailored for dual-arm manipulation. CALVIN (Mees et al., 2022) presents a language-conditioned manipulation benchmark, while MuBIE (Nazarczuk et al., 2025) enhances realism by jointly providing high-fidelity visual rendering and accurate physical simulation for manipulation tasks. Despite these advances, mainstream simulation pipelines largely omit tactile sensing, thereby limiting the ability of models to learn fine-grained physical interactions. In contrast to these benchmarks, Tabero is the first to jointly provide scalable vision-tactile-language data generation and a standardized evaluation protocol that explicitly quantifies interaction gentleness alongside task success.

**Tactile Simulation.** Tactile simulation remains a long-standing challenge in robotics simulation. Tacto (Wang et al., 2022) replaces computationally expensive finite element methods with an elastic deformation model and leverages GPU-based rendering to generate tactile images efficiently. Taxim (Si & Yuan, 2022) tackles the desynchronization between optical response and marker motion in GelSight simulation, enabling high-speed generation of dynamic tactile signals. FOTS (Zhao et al., 2024) introduces a fast calibration procedure and a low-cost simulation plugin, while TacSL (Akinola et al., 2025) proposes a parallelized tactile simulation architecture that achieves extremely high throughput. Difftactile (Si et al., 2024) provides differentiable tactile simulation, facilitating end-to-end policy opti-

mization and system identification to narrow the sim-to-real gap. TacEx (Nguyen et al., 2024) unifies tactile simulation standards within the Isaac ecosystem, resolving fragmentation across multiple simulation engines. Taccel (Li et al., 2025) overcomes the efficiency bottleneck of vision-based tactile simulation, enabling large-scale tactile learning for robots. Our work builds upon TacEx, Taxim, and FOTS, leveraging the data infrastructure of Isaac Sim to extend tactile simulation beyond single-sensor validation toward large-scale, diverse robotic manipulation task generation.

**VTLA Models.** VTLA (Zhang et al., 2025a; Hao et al., 2025) models aim to overcome weak cross-modal temporal reasoning and poor action generalization in contact-intensive tasks such as peg insertion. TA-VLA (Zhang et al., 2025b) incorporates torque as a proxy for tactile perception, addressing the lack of physical feedback in conventional VLA models and improving performance on contact-sensitive operations. OmniVTLA (Cheng et al., 2025) tackles the heterogeneity of tactile data by aligning tactile signals semantically with vision and language, thereby enhancing stability across diverse robotic grasping and manipulation scenarios. VLA-Touch (Bi et al., 2025) introduces a two-stage tactile feedback mechanism to refine task planning and execution accuracy when visual information is ambiguous, with a focus on contact-rich tasks using the Franka arm. Tactile-VLA (Huang et al., 2025) activates physical priors within VLA models, enabling tactile-driven coordination of force and position control to improve robustness in complex environments. Inspired by these approaches, we propose Tabero-VTLA, a benchmark model suite tailored to the characteristics of Tabero data, and introduce a more refined semantic–force control mechanism.

## 3. Method

### 3.1. Cross-Platform Data Reutilization

Open-source robotic manipulation datasets constitute a valuable community resource. Rather than collecting data from scratch, our goal is to construct a pipeline that transcribes datasets originally built on MuJoCo or other platforms into the Tabero platform based on Isaac Sim. This transcription enhances visual fidelity through high-quality image rendering and enriches the data with tactile information.

We begin by reconstructing task environments in Isaac Lab that closely match those of the source domains, including assets, initial object and robot poses, and success evaluation logic. However, direct migration faces two key challenges. First, the end-effector differs: we equip the robot with a gripper integrated with visuo-tactile sensors, whose geometry deviates from the original design. Second, the underlying controller varies: most source datasets rely on Operational Space Control (OSC), which is highly sensitive to physical

parameters, making naive playback prone to instability or divergence. To address this issue, we align the tool center point (TCP) of the end-effector by adjusting the base pose of the robot arm and use a high-gain PD joint controller during trajectory replay to minimize cumulative tracking error.

In the end, the robotic arm equipped with a tactile gripper achieved an acceptable task success rate during data collection, which we also refer to as the data retention rate.

### 3.2. Cross-Modal Data Acquisition

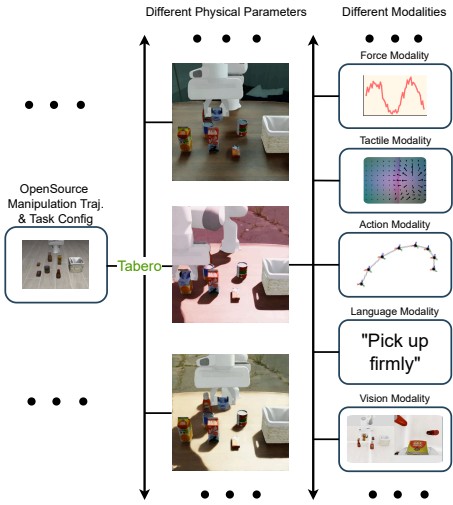

*Figure 2.* **Overview of the High-Fidelity Multimodal Data Generation Pipeline.** We take open-source trajectories and task setups originally developed for other platforms, such as MuJoCo, and replay them in our Tabero system. Tabero produces high-quality, temporally aligned data across multiple modalities, including vision, touch, and robot proprioception.

Leveraging the GPU-accelerated parallel rendering capabilities of Isaac Lab, we build a real-time, synchronized multimodal data acquisition system (Fig. 2) that captures diverse sensory streams at high fidelity. Visual information is obtained from two camera configurations: a wrist-mounted camera and a third-person view camera, both providing synchronized RGB-D data. Tactile observations are produced by a simulated GelSight (Yuan et al., 2017) sensor with a resolution of $320 \times 240$, providing both RGB tactile images and an $11 \times 9$ marker grid. In subsequent experiments, we analyze marker-free tactile images and image-free marker matrices independently.

**Tactile Image.** Tactile images are generated using the Taxim framework (Si & Yuan, 2022), which simulates illumination changes in tactile sensors via photometric stereo (Johnson & Adelson, 2009). This produces high-

resolution images encoding fine contact surface geometry.

**Marker Displacement Field.** Following the FOTS approach, we model the displacement field of surface markers on an elastic tactile sensor. This field provides a raw but informative signal for estimating shear forces and torques. Moreover, this representation is compatible with a wide range of tactile sensor technologies, including piezoelectric, capacitive, and magnetoresistive designs, which makes our approach hardware-agnostic.

**Contact Force.** Ground-truth 6D contact forces come from simulated tactile sensors on the left and right fingertips. The physics engine provides the force at each contact patch, denoted $\mathbf{F}_{\text{left}}$ and $\mathbf{F}_{\text{right}}$. We compute grip force from their normal components and use their sum as the applied force for supervision during training and evaluation.

All cameras are rendered in parallel using tiled rendering, and all modalities, including visual, tactile, force, language instructions, and executed actions, are sampled synchronously at 20 Hz to produce temporally aligned multimodal observations at each time step.

## 3.3. Enriching Tactile Force Diversity

Most open-source robotic datasets use continuous actions for arm control but binary or discrete commands for the gripper, limiting fine-grained force regulation and leading to narrow force distributions within tasks. To address this, we execute the same task with varied force magnitudes by modulating low-level gripper controller parameters.

In simulation, grippers are typically controlled via impedance control. For a single-DOF gripper with width $p$, the commanded force is:

$$F^{\text{cmd}} = K_p(p^{\text{target}} - p) + K_d(\dot{p}^{\text{target}} - \dot{p}). \quad (1)$$

At steady state after contact, velocities vanish and the static grip force approximates $F_{\text{static}} \approx K_p \cdot \delta$, where $\delta = p - p^{\text{target}} > 0$ is the penetration depth. Thus, varying $K_p$ directly scales the steady-state force under identical high-level commands. During initial contact, however, the impact force is dominated by damping: $F_{\text{impact}} \propto K_d \cdot |\dot{p}|$, so adjusting $K_d$ diversifies transient forces.

We leverage this by collecting trajectories with different $(K_p, K_d)$ settings to simulate both gentle and firm grasps. The resulting interactions produce distinct tactile force patterns on the left and right fingertips, captured as $\mathbf{F}_{\text{left}}$ and $\mathbf{F}_{\text{right}}$. Language instructions are augmented with adverbs such as "gently" or "softly" for low-force interactions and "firmly" or "tightly" for high-force ones, aligning semantics with the measured fingertip forces. We also log continuous gripper aperture $p$ instead of discrete open/close signals, en-

suring compatibility between action space and force dynamics. This pipeline enables flexible generation of arbitrary force profiles for any task.

## 3.4. Tabero-VTLA

Tabero-VTLA is trained on the Tabero dataset. We adopt the tactile marker motion field as the default tactile modality, as it directly encodes the magnitude and direction of normal force, shear force, and torque, and generalizes across piezoelectric, magnetic, and vision-based tactile sensors (Xue et al., 2025). To integrate this tactile signal into the VLA foundation model, we introduce a tactile tokenizer that maps tactile inputs into conditional tokens. The high-level policy then jointly predicts future gripper poses and fingertip force setpoints based on vision, language, and touch. These commands are tracked by a fixed admittance-based low-level controller, which provides physical compliance; the policy itself does not implement compliance but learns to reason about appropriate interaction forces given the task context. Building on the Pi0 infrastructure and leveraging flow matching, our approach enables continuous prediction of both pose and force. Below, we detail the tactile tokenizer and loss function, and also compare alternative tactile injection strategies inspired by prior work.

**Force-Field Tokenizer.** It processes tactile marker displacement fields relative to the initial rest state of the sensor. The input consists of $H + 1$ frames: the first frame captures the undeformed marker layout, and the subsequent $H$ frames record 2D marker positions in the sensor's local plane, resulting in an array $M \in \mathbb{R}^{(H+1)\times N \times 2}$. Here, $N$ denotes the number of markers, and the 2D element denotes the marker's planar coordinates. A lightweight Temporal Convolutional Network (TCN) encodes this spatiotemporal sequence into tokens for integration into the transformer backbone. Detailed architecture and hyperparameters are provided in Appendix A.

**Tactile-Image Adapter.** We construct a single composite tactile image by arranging the $H$ historical frames from the left finger and the $H$ frames from the right finger into a unified spatial layout. This image, encoding the full tactile history of both fingertips, is processed by the same visual encoder used for RGB-D observations. Its features then interact with visual features via cross-attention in the transformer, enabling joint reasoning over contact history and scene geometry. Implementation details and parameter settings are provided in the Appendix A.

**Force Tokenizer.** We use the 3D force vectors measured at the left and right fingertips, denoted $\mathbf{F}_{\text{left}}$ and $\mathbf{F}_{\text{right}}$, as raw inputs, resulting in a 6-dimensional force signal. Although these fingertip forces can be decomposed to recover

the full 6D interaction wrench on the object, we find it more effective to directly feed the concatenated 6D vector into a multilayer perceptron (MLP) to obtain a compact latent representation of contact dynamics. This representation is injected into the model alongside other modalities to enhance physical awareness. Specific network configurations and training parameters are provided in the Appendix A.

**Force Supervision** Flow matching naturally supports continuous force prediction, motivating our weighted loss design. The action vector $a$ includes both position commands and target forces derived from tactile sensing. For a batch $b$ and time $t \sim \mathcal{U}(0, 1)$, we define the interpolated input $x_t = (1 - t)\epsilon + ta$ and target velocity $u_t = a - \epsilon$, where $\epsilon \sim \mathcal{N}(0, I)$. The prediction error $e = v_t - u_t$ is split into action and force components, $[e^{(\text{act})}, e^{(\text{force})}]$. We upweight the force dimensions by a factor $\lambda_{\text{force}} > 0$, while keeping other dimensions at unit weight. The final loss is:

$$\mathcal{L} = \frac{1}{D_{\text{act}}} \sum_{d=1}^{D_{\text{act}}} w_d (e_d)^2, \qquad (2)$$

where $w_d = \lambda_{\text{force}}$ for dimensions corresponding to predicted forces, and $w_d = 1$ otherwise, with $\lambda_{\text{force}} > 0$.

### 3.5. Decoupled Force–Position Hybrid Controller

We build upon our force control design based on Tactile-VLA and further extend it to enable precise grasp force control: we decouple the grip force from the translational force applied to the object and establish separate control models for closed-loop regulation of each. We measure 3D fingertip forces $\mathbf{F}_{\text{left}}$ and $\mathbf{F}_{\text{right}}$ via tactile sensors, expressed in a finger-local frame where the $z$-axis aligns with the gripping direction and corresponds to the normal contact force. The grip and applied forces are computed as:

$$F_{\text{grip}} = 2 \cdot \min\left(|F_{\text{left},z}|, |F_{\text{right},z}|\right), \qquad (3)$$
$$\mathbf{F}_{\text{applied}} = -(\mathbf{F}_{\text{left}} + \mathbf{F}_{\text{right}}), \qquad (4)$$

where $F_{\text{left},z}$ and $F_{\text{right},z}$ denote the $z$-components of the left and right finger forces, respectively.

**Applied Force Control** Let $\Sigma_B$ denote the robot base frame and $\Sigma_C$ the local contact frame. The policy outputs a desired end-effector pose $\mathbf{P}^{\text{pred}} \in SE(3)$ and a target applied force $\mathbf{F}_{\text{applied}}^{\text{target}}$ (expressed in $\Sigma_C$). The measured contact force is denoted $\mathbf{F}_{\text{applied}}^{\text{meas}}$ (expressed in $\Sigma_B$). To introduce compliance, we apply an admittance-based position correction. The hybrid target position command $\mathbf{p}^{\text{cmd}}$ is given by:

$$\mathbf{p}^{\text{cmd}} = \mathbf{P}^{\text{pred}} + \mathbf{K}_P^{\text{adm}} \left( \mathbf{R}_C^B \mathbf{F}_{\text{applied}}^{\text{target}} - \mathbf{F}_{\text{applied}}^{\text{meas}} \right), \qquad (5)$$

where $\mathbf{R}_C^B \in SO(3)$ rotates vectors from the contact frame to the base frame, and $\mathbf{K}_P^{\text{adm}} \in \mathbb{R}^{3 \times 3}$ is the admittance

gain matrix. The final joint velocity command $\dot{\mathbf{q}}$ is then computed via differential inverse kinematics.

**Grip Force Control** To overcome the limitation of position control in precise force regulation, we implement a grip force feedback loop. The target grip force is computed from the predicted grip force $F_{\text{grip}}^{\text{pred}}$ as

$$F_{\text{grip}}^{\text{target}} = (1 + k_{\text{ff}}) F_{\text{grip}}^{\text{pred}}, \quad k_{\text{ff}} > 0, \qquad (6)$$

where $k_{\text{ff}}$ is a feedforward gain. The measured grip force $\tilde{F}_{\text{grip}}$ is exponentially smoothed. Let $p$ denote the gripper width. We apply a correction $\Delta p$ only when $|\Delta p| \geq \text{dz}$, where $\text{dz} > 0$ is a deadzone threshold accounting for gripper imprecision:

$$\Delta p = k_p^{\text{adm}} \cdot (F_{\text{grip}}^{\text{target}} - \tilde{F}_{\text{grip}}), \qquad (7)$$

with $k_p^{\text{adm}}$ an admittance gain. The final gripper command is

$$p^{\text{cmd}} = \text{clip}(p^{\text{pred}} + \Delta p, 0, p_{\text{max}}), \qquad (8)$$

where $p^{\text{pred}}$ is the policy-predicted gripper width. This allows the policy to specify a desired grip force while the controller achieves accurate tracking via feedforward and force feedback.

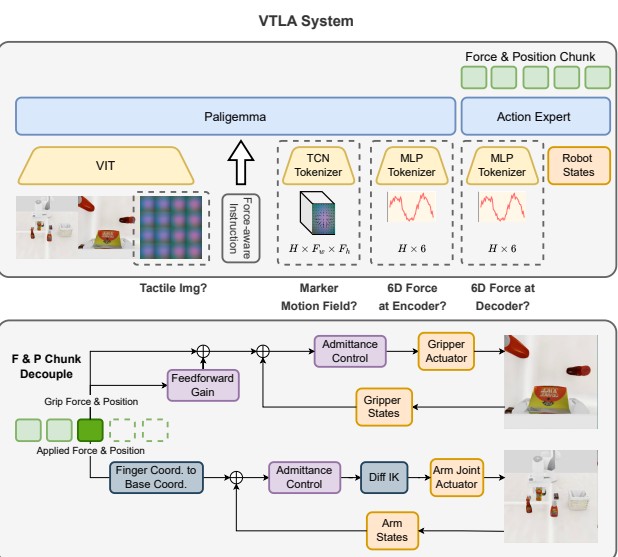

*Figure 3.* **Tabero-VTLA system overview.** VTLA system: tactile inputs are encoded by specialized modules and fused with vision and language. Real-time force feedback system: the policy predicts force-position commands, which a decoupled low-level controller tracks to achieve compliant interaction.

### 3.6. Metrics Beyond Success Rate

While existing evaluation protocols for robotic foundation models typically rely solely on task success rate as the primary metric, we regard this as an outcome-oriented assessment that overlooks critical aspects of physical interaction.

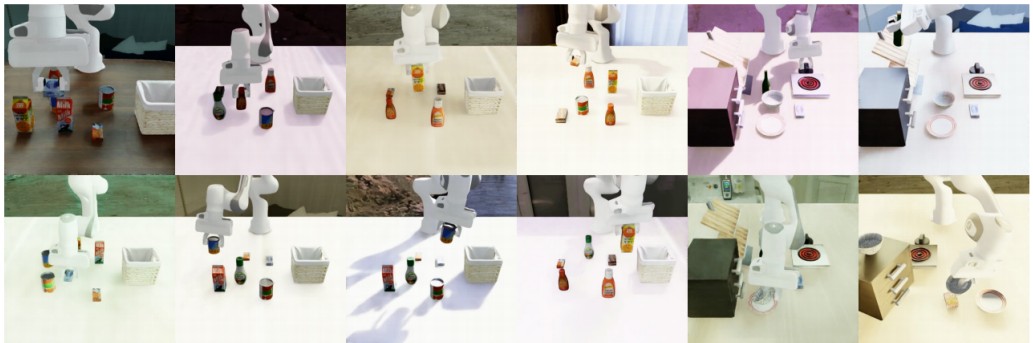

*Figure 4.* **Tabero Simulation Platform.** Tabero replicates the LIBERO task environments, enables data reuse, enhances the visual fidelity of simulated data, and makes it possible to obtain high-quality tactile modalities.

To address this limitation, we introduce a set of process-aware metrics that quantify the quality of physical interaction during task execution: **Maximum Transient Grip Force (MG).** The average of the top 5% grip force values over an episode, capturing peak grasping effort and transient spikes that may indicate aggressive or damaging behavior. **Average Grip Force (AG).** The mean grip force during contact, reflecting the nominal force used in stable grasping. **Maximum Transient Applied Force (MA).** The average of the top 5% magnitudes of the applied force, characterizing extreme interaction events such as impacts. **Average Applied Force (AA).** The mean applied force magnitude during contact, measuring overall interaction intensity.

Together, these metrics enable a more nuanced and physically grounded evaluation of robot policies, moving beyond binary success to assess safety and interaction quality.

## 4. Experiments

### 4.1. Cross-Platform Data Validation

We validate the fidelity of data migration from MuJoCo to Isaac Lab by evaluating both task success rates and distributional consistency. Specifically, we select four subtasks from the LIBERO benchmark suite and compare the success rates of the original MuJoCo-based dataset with those of our replayed version in Isaac Lab. Figure 4 illustrates the range of task scenarios our simulation platform can provide. In subsequent experiments, we only used a subset of these configurations and did not randomize the visual modality.

When using the same robot kinematics and control policy as in the original dataset, our baseline configuration yields a success rate distribution that closely matches that reported in OpenVLA (Kim et al., 2025b). However, replacing the standard end effector with a Franka arm equipped with a tactile sensor integrated gripper introduces mechanical differences that lead to a measurable drop in success rates.

This degradation is especially pronounced in tasks requiring delicate manipulation, where lower grip forces strongly correlate with reduced success. The results shown in tab. 1 highlight the sensitivity of contact-rich tasks to end-effector design and force regulation.

*Table 1.* Cross-platform data validation: Task success rates across four LIBERO subtasks. We compare the original MuJoCo dataset, our replay in Isaac Lab with identical robot configuration (denoted to Isaac), and our modified setup with a tactile-equipped Franka gripper (denoted to T-100,T-25 and T-10).

| TASK | MUJOCO | ISAAC | T-100 | T-25 | T-10 |
|------|--------|-------|-------|------|------|
| SPATIAL | 0.86 | 0.83 | 0.42 | 0.24 | 0.07 |
| OBJECT | 0.91 | 0.77 | 0.84 | 0.87 | 0.73 |
| GOAL | 0.76 | 0.78 | 0.55 | 0.44 | 0.30 |
| 10 | 0.86 | 0.66 | 0.60 | 0.47 | 0.32 |
| AVERAGE | 0.85 | 0.76 | 0.60 | 0.50 | 0.36 |

### 4.2. Tactile Data Diversity Analysis

We verify whether our data collection framework effectively expands the distribution of interaction forces. We compare a baseline using binary gripper control against our approach, which explicitly sets different force parameters during execution, the results are shown in fig. 5. Specifically, we define 100% grip force as "strong", modifying the original instruction with the adverb "firmly" and "tightly", and set 25% and 10% as "light" conditions, annotated with the adverb "gently" and "softly". In our experiments, we set $K_p \in \{2000, 500, 200\}$ N/m and $K_d \in \{100, 25, 10\}$ N·s/m to correspond to 100%, 25%, and 10% force levels, respectively, where the 100% setting follows the default parameters for the Franka robot in Isaac Sim. For the same task, we present tactile images, force fields, and contact force measurements at comparable contact stages under these distinct force settings in fig. 6, demonstrating clear multimodal variations aligned with the intended interaction intensity.

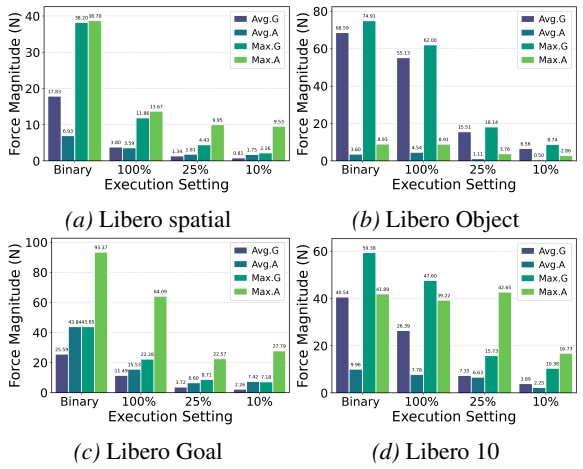

*(a) Libero spatial*      *(b) Libero Object*

*(c) Libero Goal*      *(d) Libero 10*

*Figure 5.* **Force Distribution Across Different Task Suites and Force Control Modes.** The force distribution charts show the applied forces under various control modes across different task suites. "Binary" represents the binarized control commands applied by a non-tactile gripper during tasks. "100%", "25%", and "10%" indicate the force distributions when using a tactile gripper under different force settings. The force magnitudes are determined by environmental physical parameters, with "100%" force settings matching those of the non-tactile gripper.

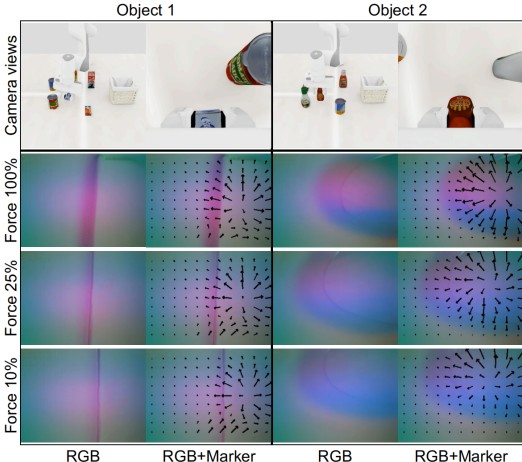

*Figure 6.* **Different force magnitudes of tactile images and corresponding camera images are illustrated.** The left two columns in the figure represent the first category of objects while the right two columns represent the second category of objects. RGB denotes the simulated tactile RGB image, and RGB+Marker indicates the effect of overlaying the tactile marker motion field simulated image with the RGB image. In the camera views, the left view is from the third-person camera and the right view is from the wrist-mounted camera.

Furthermore, we observe that under the extreme 10% force setting, the retention rate of most tasks is relatively low. All task retention results are presented in Appendix D. Therefore, in subsequent ablation tests, we constructed a Tabero subset to analyze the policy's performance under extreme conditions. This subset includes 9 tasks from the Object dataset, each executed under two force conditions specified by linguistic adverbs. The composition of the dataset is presented in Appendix C. We collect three datasets over the task group, differing only in force magnitude and gripper actuation strategy. **Dataset A** uses continuous control with "gentle" forces at 25% of the corresponding "firm" level. **Dataset B** also employs continuous control but reduces the force to 10%, representing an extreme low-force regime where slippage is likely. **Dataset C** uses the same 10% force level but switches to binary open/close commands, with only discrete gripper states logged.

During evaluation, the arm and gripper operate with impedance parameters corresponding to 100% force scaling. The Tabero-VTLA policy predicts target contact forces conditioned on language and visual inputs; these targets are tracked by a closed-loop admittance controller that modulates impedance in real time. We adapt a base VLA model using LoRA to incorporate tactile marker fields (Dataset A and B), while a vision–language-only variant is trained on Dataset C for ablation. As shown in Tab. 2, removing tactile feedback leads to complete failure in force modulation, highlighting its critical role in gentle manipulation. Furthermore, the sharp drop in success rate from 25% to 10%

force scaling reflects the increased difficulty of maintaining stable contact under ultra-gentle constraints, underscoring that Dataset B represents a substantially more challenging regime for gentleness-aware policies.

*Table 2.* Training results on three datasets

| METRIC | A 100% / 25% | B 100% / 10% | C 100% / 10% |
|---|---|---|---|
| SR | 0.87 / 0.79 | 0.86 / 0.52 | 0.86 / 0.87 |
| AG | 31.3 / **8.5** | 32.4 / **3.7** | 35.8 / 34.6 |
| MG | 61.3 / 19.9 | 62.7 / 11.4 | 65.7 / 65.0 |

### 4.3. Effectiveness of Hybrid Controller

To isolate the contribution of the low-level controller from the high-level policy, we evaluate on **Dataset A**, where task success is largely insensitive to force variations. The hyperparameters of our controller are presented in Appendix B. We conduct four ablation studies on the gripper controller: (a) full force with hybrid control, (b) reduced force with hybrid control, (c) reduced force without feedforward term, and (d) reduced force without admittance component. Results in fig. 7 demonstrate that both feedforward and admittance terms are essential for accurate force tracking.

For the arm controller, we focus only on gripper force feedback, as the applied interaction forces in our tasks are small and have limited influence on arm motion. Consequently, we highlight the effectiveness of tactile-based grip force regulation, which is critical for successful grasping.

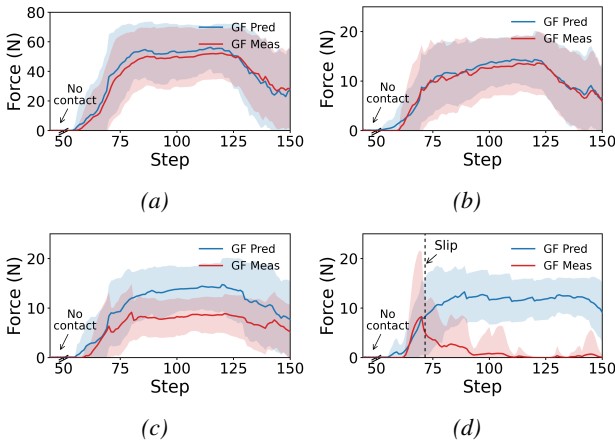

*Figure 7.* **Ablation study on gripper force control.** GF stands for gripper force. In Tabero Object task 1, the predicted force is shown in blue and the measured force in red: (a) 100% force, (b) 25% force, (c) 25% force without feedforward term, and (d) 25% force without admittance control. Slip stands for object dropping.

### 4.4. Ablation and Comparison of VTLA

To compare and conduct ablation experiments on different tactile injection methods and force control strategies, we evaluate tasks from **Dataset B**. Among them, Force E+FS is designed based on Huang et al. (2025), IMG is designed based on Zhang et al. (2025a), and Force D+FS is designed based on Zhang et al. (2025b). All models, excluding the ablation architectures, were fine-tuned via LoRA with an identical set of hyperparameters, detailed parameters reported in the Appendix A.

*Table 3.* Ablation study on tactile modalities.

| MODELS | F SR | G SR | F AG | G AG |
|---|---|---|---|---|
| NONE | 0.00 | 0.00 | 0.0 | 0.0 |
| IMG | 0.37 | 0.01 | 3.0 | 1.1 |
| FIELD | 0.40 | 0.01 | 2.9 | 2.0 |
| FORCE E | 0.40 | 0.01 | 2.5 | 1.8 |
| FS | 0.82 | 0.45 | 30.4 | 3.1 |
| FORCE D+FS | 0.82 | 0.31 | 28.5 | 3.3 |
| FORCE E+FS | 0.84 | **0.49** | 30.3 | 3.4 |
| IMG+FS | 0.87 | **0.48** | 30.6 | 3.6 |
| FIELD+FS | 0.86 | **0.52** | 32.4 | 3.7 |

Table Notes: F SR/G SR = Success rates under Firm/Gentle force; F AG/G AG = Average grip force under Firm/Gentle force. None = No tactile input; Img = Tactile image input; Field = Force field input; Force E = Force input via MLP encoder; Force D = Force input via decoder; FS = Force supervision loss enabled.

Table 3 shows that without tactile input the baseline control policy fails completely. This confirms that gentle grasping requires not only compliant actuation but also sensory awareness of interaction forces. When tactile tokens such as images or force fields are provided, the policy gains basic force modulation ability and achieves nontrivial success. Adding explicit force supervision enables precise force prediction and substantially improves performance under gentle conditions. The best results come from combining rich tactile representations like marker fields with force supervision, highlighting their complementary roles in gentleness-aware manipulation.

### 4.5. Semantic Force Generalization

*Table 4.* Semantic-force understanding: Average contact force (N) under different linguistic adverbs. Includes both in-domain and out-of-domain (OOD) adverbs. The model is trained on dataset B.

| VERB | SR | AG | AA | MG | MA |
|---|---|---|---|---|---|
| *firmly, tightly* | 0.86 | 32.4 | 1.5 | 62.7 | 6.9 |
| *gently, softly* | 0.52 | 3.7 | 0.4 | 11.4 | 5.1 |
| *lightly* (OOD) | 0.61 | 14.5 | 0.8 | 40.8 | 7.6 |
| *forcefully* (OOD) | 0.64 | 18.1 | 0.9 | 49.7 | 6.3 |

We evaluate how well the Tabero-VTLA model with force-field tactile injection adjusts grip force in response to linguistic adverbs; results are in Table 4. Appendix E presents the generalization test results for the image and force injection methods. With force field inputs and force supervision, Tabero-VTLA applies much higher forces for "firmly" or "tightly" than for "gently" or "softly", while maintaining strong task success. For unseen adverbs such as "lightly" and "forcefully", the model produces intermediate forces that align with their meanings, though success rates decrease. This indicates some zero-shot compositional understanding of force-related language, but also shows room for improvement in generalizing semantic-force mappings.

## 5. Conclusions

We present Tabero, a framework for evaluating and enabling gentler language-conditioned manipulation through scalable tactile data generation and a standardized gentleness-aware evaluation protocol. Our Tabero-VTLA model suite demonstrates that tactile feedback can be effectively integrated into existing VLA architectures using a force-position command interface, significantly reducing interaction forces without sacrificing task performance. This work provides a practical pathway toward safer and more dexterous robotic interaction in contact-rich environments.

**Limitations.** Nevertheless, Our current framework does not jointly optimize for both task success and minimal interaction force. In ultra-gentle regimes, there exists an inherent trade-off between gentleness and reliability. Future work could explore reinforcement learning to balance these objectives. We are also developing a real-world force–position hybrid data collection system to enable robust deployment of VTLA models in physical environments.

## Impact Statement

This paper presents work whose goal is to advance the field of Robot Learning. There are many potential societal consequences of our work, none which we feel must be specifically highlighted here.

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

## A. Hyperparameters

The following table (Tab.5, Tab.6) presents some hyperparameters of the Tabero VTLA.

*Table 5.* Common training hyperparameters for Tabero.

| PARAMETER | VALUE |
|---|---|
| MODEL FAMILY | PI0 (JAX) |
| ACTION_DIM (PADDED) | 32 |
| EFFECTIVE_ACTION_DIM (SEMANTIC) | 13 |
| TACTILE_DIM (FORCE-SLOT DIM) | 6 |
| TABERO TACTILE HISTORY (FRAMES) | 8 |
| LR SCHEDULE | COSINEDECAY |
| PEAK LR | 2.5E-5 |
| DECAY LR | 2.5E-6 |
| TRAIN STEPS | 50,000 |
| BATCH_SIZE | 32 |

*Table 6.* Simplified hyperparameters.

| CONFIG | VISUAL | TACTILE STREAMS | TACTILE TOKENIZER | LOSS WEIGHT |
|---|---|---|---|---|
| IMG+FS | 3 (W/ TACTILE) | N/A | N/A | 0.1 |
| IMG | 3 (W/ TACTILE) | N/A | N/A | 0.0 |
| FORCE D + FS | 2 | SUFFIX | MLP | 0.1 |
| FORCE E | 2 | PREFIX | MLP | 0.0 |
| FORCE E + FS | 2 | PREFIX | MLP | 0.1 |
| FIELD FS | 2 | PREFIX | TCN | 0.1 |
| FIELD | 2 | PREFIX | TCN | 0.0 |
| FS | 2 | N/A | N/A | 0.1 |
| NONE | 2 | N/A | N/A | 0.0 |

Table 7 and table 8 present some hyperparameters of the tactile tokenizer.

*Table 7.* Hyperparameters of MLP tactile tokenizer

| PARAMETER | VALUE / CONSTRAINT |
|---|---|
| EXPERT WIDTH ($W$) | SUFFIX: 1024;  PREFIX: 2048 |
| HIDDEN DIM | $2 \times W$ |
| NUM LAYERS | 2 LINEAR |
| ACTIVATION | SWISH |
| INPUT DIM ($in\_dim$) | $H \times 6 = 8 \times 6 = 48$ |

## B. Controller Hyperparameters

The parameters of the decoupled force-position hybrid controller are presented in Table 9.

## C. Task Configuration

In the actual dataset processing, we use two distinct adverbs to describe the high-force and low-force scenarios respectively, and randomize their injection positions at the start and end of the instructions to enhance model robustness. Table 10 presents the injection methods for one of the Tabero tasks.

All task names of the Tabero subset employed in our experiments are provided in Table 11.

*Table 8.* Hyperparameters of TCN tactile tokenizer

| PARAMETER | VALUE / CONSTRAINT |
|---|---|
| EXPERT WIDTH ($W$) | PREFIX: 2048; |
| NUM LAYERS | 2 |
| KERNEL SIZE | 3 (CAUSAL) |
| HISTORY ($H$) | 8 |
| ACTIVATION | SWISH |
| INPUT DIM | $11 \times 9 \times 2 \times 9 \times 2 = 3564$ |

*Table 9.* Controller parameters (Hybrid+Tactile configuration).

| PARAMETER | VALUE |
|---|---|
| GRIP.FF.K | 0.6 |
| GRIP.FF.DEADZONE | 1.0 |
| POS.KP(X,Y,Z) | (-0.0001, -0.0001, -0.0001) |
| GRIP.KP | 0.0008 |
| GRIP.DEADZONE | 0.25 |
| FORCE.FILTER.ALPHA | 0.2 |
| IK.COMMAND.TYPE | POSE |
| IK.IK.METHOD | DLS |
| IK.SCALE | 1.0 |
| IK.OFFSET.POS | [0.0, 0.0, 0.1034] |
| ARM.PD.SHOULDER(STIFFNESS, DAMPING) | (8000.0, 800.0) |
| ARM.PD.FOREARM(STIFFNESS, DAMPING) | (8000.0, 800.0) |
| GRIPPER.PD.PANDA.HAND(STIFFNESS, DAMPING) | (2000.0, 100.0) |

## D. Data Retention Results

In this section, Table 12 presents the retention rate of the gripper with a tactile sensor at 100% force, Table 13 at 10% force, and Table 14 at 25% force. We define the data retention rate as the ratio of the data with successfully completed tasks during the replay process to the total data volume, which reflects the stability of the data to a certain extent. Mathematically, it is expressed as:

$$R = \frac{N_s}{N_t} \times 100\% \tag{9}$$

where $R$ denotes the data retention rate, $N_s$ represents the amount of data for which the task is successfully completed in the replay process, and $N_t$ is the total data volume.

## E. Additional Generalization Test

We present the generalization test results for the tactile image injection and tactile force injection modes in Table 16 and Table 15.

*Table 10.* Adverb injection methods within the same task.

| TASK_INDEX | TASK DESCRIPTION |
|---|---|
| 0 | ***tightly*** *pick up the alphabet soup and place it in the basket* |
| 1 | ***firmly*** *pick up the alphabet soup and place it in the basket* |
| 2 | *pick up the alphabet soup and place it in the basket* ***tightly*** |
| 3 | *pick up the alphabet soup and place it in the basket* ***firmly*** |
| 4 | ***softly*** *pick up the alphabet soup and place it in the basket* |
| 5 | *pick up the alphabet soup and place it in the basket* ***gently*** |
| 6 | *pick up the alphabet soup and place it in the basket* ***softly*** |
| 7 | ***gently****pick up the alphabet soup and place it in the basket* |

*Table 11.* Tabero subset

| TASK ID | TASK DESCRIPTION |
|---|---|
| 0 | *pick up the alphabet soup and place it in the basket* |
| 1 | *pick up the cream cheese and place it in the basket* |
| 2 | *pick up the salad dressing and place it in the basket* |
| 3 | *pick up the bbq sauce and place it in the basket* |
| 4 | *pick up the tomato sauce and place it in the basket* |
| 5 | *pick up the butter and place it in the basket* |
| 6 | *pick up the milk and place it in the basket* |
| 7 | *pick up the chocolate pudding and place it in the basket* |
| 8 | *pick up the orange juice and place it in the basket* |

*Table 12.* Task completion performance and force data of the tactile gripper at 100% tactile force

| TASK | $n$ | RATIO (%) | MAX STEPS | AG | AA | MG | MA |
|---|---|---|---|---|---|---|---|
| 10_TASK0 | 20 | 40.0 | 343 | 71.216 | 7.714 | 92.178 | 20.567 |
| 10_TASK1 | 37 | 74.0 | 322 | 51.621 | 1.629 | 67.509 | 6.456 |
| 10_TASK2 | 37 | 74.0 | 340 | 7.179 | 41.725 | 35.092 | 241.123 |
| 10_TASK3 | 36 | 72.0 | 317 | 3.777 | 2.272 | 12.111 | 13.664 |
| 10_TASK4 | 25 | 50.0 | 298 | 5.112 | 2.055 | 9.767 | 11.047 |
| 10_TASK5 | 35 | 70.0 | 244 | 10.544 | 5.542 | 17.715 | 21.959 |
| 10_TASK6 | 26 | 52.0 | 329 | 24.796 | 2.429 | 92.066 | 10.938 |
| 10_TASK7 | 37 | 74.0 | 312 | 59.914 | 6.566 | 84.572 | 20.859 |
| 10_TASK8 | 20 | 40.0 | 459 | 16.167 | 4.168 | 45.458 | 20.258 |
| 10_TASK9 | 28 | 56.0 | 449 | 13.543 | 3.697 | 19.518 | 25.310 |
| GOAL_TASK0 | 24 | 48.0 | 196 | 0.000 | 15.926 | 0.000 | 27.606 |
| GOAL_TASK1 | 23 | 46.0 | 119 | 3.670 | 3.495 | 13.624 | 15.170 |
| GOAL_TASK2 | 41 | 82.0 | 146 | 31.829 | 6.010 | 38.518 | 12.699 |
| GOAL_TASK3 | 2 | 4.0 | 184 | 3.202 | 3.182 | 7.255 | 19.636 |
| GOAL_TASK4 | 19 | 38.0 | 123 | 3.363 | 4.137 | 11.705 | 18.866 |
| GOAL_TASK5 | 22 | 44.0 | 178 | 0.091 | 3.619 | 1.498 | 11.900 |
| GOAL_TASK6 | 34 | 68.0 | 169 | 56.255 | 2.730 | 83.899 | 12.032 |
| GOAL_TASK7 | 45 | 90.0 | 119 | 8.906 | 106.678 | 43.984 | 490.124 |
| GOAL_TASK8 | 21 | 42.0 | 126 | 3.912 | 4.390 | 12.752 | 17.756 |
| GOAL_TASK9 | 43 | 86.0 | 258 | 3.671 | 5.162 | 9.575 | 15.134 |
| OBJECT_TASK0 | 45 | 90.0 | 196 | 89.895 | 4.905 | 95.571 | 12.011 |
| OBJECT_TASK1 | 48 | 96.0 | 187 | 55.012 | 1.889 | 63.747 | 4.496 |
| OBJECT_TASK2 | 44 | 88.0 | 170 | 37.727 | 3.930 | 41.180 | 8.158 |
| OBJECT_TASK3 | 36 | 72.0 | 222 | 22.649 | 3.139 | 26.134 | 7.497 |
| OBJECT_TASK4 | 45 | 90.0 | 248 | 34.886 | 5.970 | 39.493 | 10.712 |
| OBJECT_TASK5 | 28 | 56.0 | 177 | 76.958 | 6.667 | 86.918 | 11.584 |
| OBJECT_TASK6 | 41 | 82.0 | 207 | 49.276 | 1.954 | 58.672 | 4.597 |
| OBJECT_TASK7 | 43 | 86.0 | 175 | 60.649 | 5.810 | 66.240 | 11.439 |
| OBJECT_TASK8 | 44 | 88.0 | 174 | 68.241 | 4.071 | 78.576 | 5.543 |
| OBJECT_TASK9 | 46 | 92.0 | 254 | 55.968 | 7.056 | 63.477 | 13.038 |
| SPATIAL_TASK0 | 20 | 40.0 | 168 | 4.834 | 5.200 | 16.239 | 19.770 |
| SPATIAL_TASK1 | 29 | 58.0 | 189 | 3.447 | 3.684 | 11.151 | 15.978 |
| SPATIAL_TASK2 | 6 | 12.0 | 164 | 3.647 | 4.598 | 14.820 | 16.432 |
| SPATIAL_TASK3 | 25 | 50.0 | 119 | 3.105 | 4.202 | 9.499 | 16.272 |
| SPATIAL_TASK4 | 18 | 36.0 | 190 | 4.250 | 3.280 | 10.766 | 10.565 |
| SPATIAL_TASK5 | 14 | 28.0 | 142 | 4.366 | 2.690 | 12.019 | 9.047 |
| SPATIAL_TASK6 | 13 | 26.0 | 159 | 3.316 | 2.429 | 11.466 | 10.947 |
| SPATIAL_TASK7 | 29 | 58.0 | 180 | 3.497 | 3.324 | 11.409 | 12.271 |
| SPATIAL_TASK8 | 34 | 68.0 | 157 | 3.856 | 3.432 | 11.996 | 13.236 |
| SPATIAL_TASK9 | 20 | 40.0 | 193 | 3.687 | 3.051 | 9.253 | 12.185 |

*Table 13.* Task completion performance and force data of the tactile gripper at 10% tactile force

| TASK | $n$ | RATIO (%) | MAX STEPS | AG | AA | MG | MA |
|---|---|---|---|---|---|---|---|
| 10_TASK0 | 14 | 28.0 | 388 | 7.934 | 1.211 | 13.115 | 7.970 |
| 10_TASK1 | 39 | 78.0 | 322 | 5.943 | 0.429 | 10.366 | 3.794 |
| 10_TASK2 | 3 | 6.0 | 242 | 0.583 | 8.170 | 5.660 | 59.902 |
| 10_TASK3 | 8 | 16.0 | 269 | 0.771 | 1.814 | 2.059 | 11.944 |
| 10_TASK4 | 17 | 34.0 | 298 | 0.940 | 1.080 | 2.242 | 7.509 |
| 10_TASK5 | 39 | 78.0 | 259 | 2.782 | 1.636 | 4.168 | 19.098 |
| 10_TASK6 | 13 | 26.0 | 276 | 4.948 | 0.786 | 23.483 | 4.222 |
| 10_TASK7 | 23 | 46.0 | 312 | 7.316 | 1.797 | 13.288 | 15.329 |
| 10_TASK8 | 0 | 0.0 | 0 | – | – | – | – |
| 10_TASK9 | 3 | 6.0 | 395 | 3.767 | 3.341 | 19.068 | 20.798 |
| GOAL_TASK0 | 22 | 44.0 | 196 | 0.000 | 12.768 | 0.000 | 20.558 |
| GOAL_TASK1 | 4 | 8.0 | 115 | 0.711 | 1.563 | 2.549 | 10.536 |
| GOAL_TASK2 | 24 | 48.0 | 137 | 4.939 | 0.842 | 5.669 | 2.448 |
| GOAL_TASK3 | 0 | 0.0 | 0 | – | – | – | – |
| GOAL_TASK4 | 7 | 14.0 | 108 | 0.751 | 4.334 | 2.835 | 23.310 |
| GOAL_TASK5 | 18 | 36.0 | 178 | 0.032 | 3.637 | 1.056 | 11.691 |
| GOAL_TASK6 | 17 | 34.0 | 136 | 10.594 | 0.962 | 38.848 | 6.497 |
| GOAL_TASK7 | 37 | 74.0 | 119 | 1.059 | 40.585 | 8.926 | 168.978 |
| GOAL_TASK8 | 4 | 8.0 | 112 | 0.601 | 0.907 | 1.303 | 3.828 |
| GOAL_TASK9 | 17 | 34.0 | 184 | 1.650 | 1.163 | 3.476 | 2.226 |
| OBJECT_TASK0 | 36 | 72.0 | 196 | 9.890 | 0.803 | 13.927 | 5.568 |
| OBJECT_TASK1 | 47 | 94.0 | 187 | 6.130 | 0.236 | 7.398 | 1.740 |
| OBJECT_TASK2 | 30 | 60.0 | 170 | 5.058 | 0.442 | 6.362 | 2.912 |
| OBJECT_TASK3 | 34 | 68.0 | 222 | 3.415 | 0.290 | 4.279 | 1.468 |
| OBJECT_TASK4 | 13 | 26.0 | 239 | 4.875 | 0.636 | 6.102 | 6.659 |
| OBJECT_TASK5 | 31 | 62.0 | 180 | 9.897 | 0.342 | 11.977 | 2.250 |
| OBJECT_TASK6 | 41 | 82.0 | 207 | 6.008 | 0.155 | 9.627 | 1.327 |
| OBJECT_TASK7 | 45 | 90.0 | 175 | 6.824 | 0.844 | 8.135 | 2.587 |
| OBJECT_TASK8 | 42 | 84.0 | 174 | 7.769 | 0.226 | 12.142 | 1.152 |
| OBJECT_TASK9 | 45 | 90.0 | 254 | 5.703 | 1.001 | 7.493 | 2.907 |
| SPATIAL_TASK0 | 3 | 6.0 | 168 | 0.454 | 5.048 | 1.997 | 22.603 |
| SPATIAL_TASK1 | 8 | 16.0 | 150 | 0.896 | 1.115 | 2.284 | 7.844 |
| SPATIAL_TASK2 | 0 | 0.0 | 0 | – | – | – | – |
| SPATIAL_TASK3 | 1 | 2.0 | 115 | 0.956 | 0.630 | 1.932 | 2.567 |
| SPATIAL_TASK4 | 3 | 6.0 | 158 | 0.870 | 1.002 | 1.441 | 6.226 |
| SPATIAL_TASK5 | 5 | 10.0 | 142 | 0.905 | 1.411 | 3.110 | 7.988 |
| SPATIAL_TASK6 | 4 | 8.0 | 159 | 0.772 | 1.819 | 1.888 | 10.425 |
| SPATIAL_TASK7 | 2 | 4.0 | 180 | 0.771 | 1.065 | 1.514 | 6.294 |
| SPATIAL_TASK8 | 7 | 14.0 | 127 | 0.664 | 2.435 | 3.205 | 10.360 |
| SPATIAL_TASK9 | 3 | 6.0 | 139 | 1.036 | 1.219 | 2.029 | 11.435 |

*Table 14.* Task completion performance and force data of the tactile gripper at 25% tactile force

| TASK | $n$ | RATIO (%) | MAX STEPS | AG | AA | MG | MA |
|---|---|---|---|---|---|---|---|
| 10_TASK0 | 23 | 46.0 | 388 | 17.598 | 2.489 | 25.610 | 10.602 |
| 10_TASK1 | 41 | 82.0 | 322 | 14.410 | 0.914 | 20.463 | 6.557 |
| 10_TASK2 | 4 | 8.0 | 299 | 1.177 | 43.789 | 13.188 | 289.938 |
| 10_TASK3 | 28 | 56.0 | 291 | 1.419 | 1.676 | 4.321 | 11.437 |
| 10_TASK4 | 25 | 50.0 | 298 | 1.784 | 0.936 | 3.714 | 6.844 |
| 10_TASK5 | 35 | 70.0 | 259 | 0.714 | 2.593 | 1.883 | 15.586 |
| 10_TASK6 | 20 | 40.0 | 329 | 8.663 | 1.079 | 34.446 | 6.894 |
| 10_TASK7 | 36 | 72.0 | 312 | 15.813 | 2.463 | 25.500 | 10.698 |
| 10_TASK8 | 0 | 0.0 | 0 | – | – | – | – |
| 10_TASK9 | 23 | 46.0 | 395 | 4.530 | 3.701 | 12.427 | 25.288 |
| GOAL_TASK0 | 23 | 46.0 | 196 | 0.000 | 16.925 | 0.000 | 31.795 |
| GOAL_TASK1 | 10 | 20.0 | 115 | 1.324 | 1.696 | 4.867 | 9.737 |
| GOAL_TASK2 | 32 | 64.0 | 142 | 11.036 | 1.902 | 12.938 | 4.658 |
| GOAL_TASK3 | 1 | 2.0 | 184 | 1.465 | 0.973 | 2.463 | 10.602 |
| GOAL_TASK4 | 12 | 24.0 | 112 | 1.445 | 3.250 | 5.779 | 15.204 |
| GOAL_TASK5 | 28 | 56.0 | 215 | 0.080 | 3.249 | 1.099 | 11.181 |
| GOAL_TASK6 | 32 | 64.0 | 152 | 16.710 | 0.898 | 36.053 | 6.811 |
| GOAL_TASK7 | 40 | 80.0 | 119 | 1.665 | 32.591 | 14.429 | 118.504 |
| GOAL_TASK8 | 6 | 12.0 | 112 | 1.288 | 2.802 | 4.752 | 13.082 |
| GOAL_TASK9 | 35 | 70.0 | 258 | 2.188 | 1.684 | 4.740 | 4.129 |
| OBJECT_TASK0 | 45 | 90.0 | 196 | 23.007 | 1.656 | 26.037 | 6.328 |
| OBJECT_TASK1 | 48 | 96.0 | 187 | 15.160 | 0.437 | 17.989 | 2.404 |
| OBJECT_TASK2 | 45 | 90.0 | 170 | 11.284 | 0.960 | 12.375 | 3.249 |
| OBJECT_TASK3 | 37 | 74.0 | 222 | 4.328 | 0.881 | 5.833 | 2.933 |
| OBJECT_TASK4 | 46 | 92.0 | 251 | 11.953 | 0.761 | 13.112 | 5.234 |
| OBJECT_TASK5 | 33 | 66.0 | 184 | 23.960 | 0.904 | 26.105 | 4.838 |
| OBJECT_TASK6 | 42 | 84.0 | 207 | 14.480 | 0.350 | 19.391 | 1.880 |
| OBJECT_TASK7 | 47 | 94.0 | 196 | 17.339 | 2.042 | 19.602 | 4.226 |
| OBJECT_TASK8 | 45 | 90.0 | 174 | 18.869 | 0.705 | 23.694 | 1.473 |
| OBJECT_TASK9 | 45 | 90.0 | 254 | 14.733 | 2.392 | 17.213 | 5.050 |
| SPATIAL_TASK0 | 10 | 20.0 | 168 | 1.414 | 2.635 | 4.795 | 13.173 |
| SPATIAL_TASK1 | 20 | 40.0 | 189 | 1.428 | 1.715 | 4.673 | 9.383 |
| SPATIAL_TASK2 | 3 | 6.0 | 125 | 1.291 | 2.116 | 4.337 | 16.955 |
| SPATIAL_TASK3 | 14 | 28.0 | 171 | 1.358 | 1.643 | 5.601 | 7.007 |
| SPATIAL_TASK4 | 9 | 18.0 | 190 | 1.426 | 1.343 | 4.084 | 6.679 |
| SPATIAL_TASK5 | 13 | 26.0 | 147 | 1.792 | 1.804 | 4.202 | 8.686 |
| SPATIAL_TASK6 | 8 | 16.0 | 159 | 1.203 | 1.819 | 4.689 | 10.464 |
| SPATIAL_TASK7 | 13 | 26.0 | 180 | 1.216 | 1.178 | 3.701 | 6.233 |
| SPATIAL_TASK8 | 19 | 38.0 | 138 | 1.062 | 2.106 | 4.396 | 8.630 |
| SPATIAL_TASK9 | 13 | 26.0 | 193 | 1.232 | 1.772 | 3.792 | 12.315 |

*Table 15.* Generalization Test for Force-based Tabero-VTLA

| LANGUAGE INSTRUCTION | SR | AVG. G | AVG. A | MAX. G | MAX. A |
|---|---|---|---|---|---|
| *firmly, tightly* | 0.82 | 28.5 | 1.2 | 59.6 | 6.5 |
| *gently, softly* | 0.31 | 3.3 | 0.4 | 10.3 | 5.4 |
| *lightly* (OOD) | 0.47 | 9.5 | 0.4 | 26.8 | 4.3 |
| *forcefully* (OOD) | 0.54 | 11.0 | 0.6 | 28.5 | 5.3 |

*Table 16.* Generalization Test for Img-based Tabero-VTLA

| LANGUAGE INSTRUCTION | SR | AVG. G | AVG. A | MAX. G | MAX. A |
|---|---|---|---|---|---|
| *firmly, tightly* | 0.87 | 30.6 | 1.5 | 61.4 | 6.6 |
| *gently, softly* | 0.48 | 3.6 | 0.3 | 11.4 | 4.4 |
| *lightly* (OOD) | 0.62 | 10.6 | 0.6 | 31.9 | 5.9 |
| *forcefully* (OOD) | 0.70 | 11.8 | 0.4 | 33.5 | 4.4 |

