# OpenReview forum: "Tabero: Learning Gentle Manipulation with Closed-Loop Force Feedback from Vision, Touch, and Language"
_ICML.cc/2026/Conference — ICML 2026 regular_

### Official Review · Reviewer_tHpJ · 2026-02-24

**Soundness:** 2
**Presentation:** 2
**Significance:** 3
**Originality:** 2
**Overall Recommendation:** 3
**Confidence:** 3

**Summary:**

- Introduces Tabero, a benchmark for language-conditioned "gentle" manipulation requiring fine-grained force perception.
- Proposes a data-efficient pipeline that repurposes open-source trajectories (e.g., Libero) into visuo-tactile-language tasks using the Isaac Lab/Taxim simulation.
- Develops Tabero-VTLA, an architecture outputting force-position commands executed by an admittance-based hybrid controller.
Proposes an evaluation protocol measuring task success along with peak/average forces.

**Compliance With Llm Reviewing Policy:**

Affirmed.

**Final Justification:**

While the authors addressed several points in the rebuttal, the gap between the proposed tactile simulation and physical reality remains a primary concern.

**Key Questions For Authors:**

- What are the differences in tactile simulation quality between Taxim and TacSL?
- Is the force control loop running at the same frequency as the robot position control?
- The benchmark allows us to gather metrics about max/average applied force. Can you clarify what needs to be optimized for physically grounded evaluation? For instance, how does the benchmark capture reactivity?
- How does the metrics gather in simulation reflect the performance of the policy in real-world settings? Are the force metrics similar, scale by some known factor, or just different?
- It’s unclear what is the main message to extract from table 2.Is it that Datasets A, B, C are more challenging settings for evaluation? But it seems like they are used for training as well, based on line 380.
- In Fig5, what is the difference between binary and 100%? Shouldn’t the force profiles be similar? Also, on what tasks?
- What are the tasks used to report metrics on Table 3. What is the evaluation protocol?
- Is it possible for the system to adapt to different force profiles in the loop, via modulating the properties of the grasp via language?
- Since language is describing effectively only 3 force profiles, how would a standard VLA model perform if trained simply on "penetration depth" as a proxy for force, without explicit tactile input? Is the tactile marker field strictly necessary for these 3-bin force levels?

**Limitations:**

Yes

**Strengths And Weaknesses:**

_Strengths_

- Addresses a significant gap in current VLA models. The lack of tactile/force modalities for contact-rich tasks.
- Uses marker displacement fields as a proxy for force, allowing the model to generalize across diverse tactile sensor types (optical, capacitive, etc.). However, this is only a hypothesis since it is not backed up by experimental results.


_Weaknesses_

- Experiments are only done in simulation, which is disconnected with the main thesis of VLA systems real physical comprehension (line 48).
- Tactile dataset is generated in simulation. Although it enables tactile images that look photo-realistic, they lack precision in encoding force information in the color gradients, in particular shear. This makes the dataset somehow disconnected from tactile data in reality.
- The marker displacement field is disconnected from the elastomer deformation expected from the real sensors. In this regard, TacSL seems a more holistic approach, since it provides both image and force field in a congruent manner.
- Language adverbs to describe force interactions seem to revert the tactile modality back to a binary state (“gently” vs “tightly”).
- Results are hard to parse. Figures can improve the labels to be more descriptive and tables could use captions with the most important information to grasp from it, rather than just “results”
- Tasks need description, for instance the difference between Libero Goal and Libero 10 in Fig 5.
- Evaluation protocol is not introduced, which prevents it from making sense of the reported metrics.
- What is the statistical significance of the results reported in Table 3?
- I consider the main challenge is to achieve good performance on the gentle force setup, which it seems to be not the case. In that case, it’s unclear the advantages of adding the language qualifiers or extra inputs.

---

> ### Author Rebuttal · Authors · 2026-03-31
>
> First, we sincerely appreciate your professional suggestions in the field of tactile sensing. We clarify and address your key concerns below.
>
> ### Explanations
>
> 1. **Contributions.**
> Our core contributions focus on proposing a multi-task, multi-metric simulation benchmark and a series of novel VTLA models. The tactile simulation pipeline we used is not proposed by us; the accuracy of color gradients, elastomer deformation, and marker motion fields has been thoroughly analyzed and validated in FOTS, Taxim, and TacEX. In fact, our tactile simulation backend is fully interchangeable; for example, it can be replaced with TacSL.
>
> 2. **Disconnection from the real world.**
> In real-world settings, robotic arms and data acquisition systems are usually costly, and task data with tactile information is difficult to collect. Meanwhile, multi-task and multi-metric evaluation is challenging to implement for VTLA models in physical environments. In contrast, simulation enables us to acquire critical information that is unavailable in the real world, especially force-related evaluation metrics (Q4). We provide real-world demonstration data at the anonymous link: https://anonymous-tabero.netlify.app/, which further validates the effectiveness of our simulation. These data will be open-sourced together with the Tabero dataset.
>
> 3. **Why we did not use TacSL.**
> To the best of our knowledge, neither the TacSL nor the original Taxim paper reports a direct, standardized comparison of tactile image quality between the two simulators(Q1). The TacSL paper shows that TacSL achieves comparable tactile image quality to a Taxim baseline while substantially improving simulation speed via GPU-based physical modeling. Taxim, in turn, demonstrates good agreement with real GelSight data and Sim2Real performance in its own evaluation, but this is not under the same protocol as TacSL. In our setting, the main design factor is not Taxim-vs-TacSL image fidelity, but interface compatibility with real sensor.
>
> 4. **Tasks, benchmark, and evaluation protocol.**
> Libero Goal is a procedural manipulation task, while Libero 10 is a long-horizon sequential task. Detailed task definitions are clearly provided in the LIBERO reference (Q7). The evaluation protocol is introduced in Section 3.6 of the paper: we use task success rate plus four physical interaction metrics, with 50 test runs per task, ensuring statistical significance.
>
> ### Clarifications of Misunderstandings
>
> 1. **Binary state.**
> This is a misunderstanding of our work. The term "binary state" in our paper refers to the open/close state of the gripper. （Q6） Furthermore, we establish multiple correspondence levels between force magnitudes and adverbs, such as 20%-softly, 40%-gently, 60%-tightly, and 80%-firmly. We provide a method that allows users to freely customize, define and edit these levels at will. In this paper, our experiments mainly demonstrate that the correspondence between force and linguistic expressions can be realized effectively.
>
> 2. **Language adverbs and force regulation.**
> The model is required to modulate force magnitudes based on language adverbs, rather than only targeting gentle force levels (Q8). Furthermore, our closed-loop method can adapt to diverse force profiles.  Within an episode, linguistic commands can be modified at any time to change the force applied in the grasping behavior.
>
> ### Responses to Other Key Questions.
> Q2: The force control loop runs at the same frequency as the position control loop, enabling real-time feedback control.
>
> Q3: The core optimization objective is to enable VLA models to adjust interaction forces according to language adverbs. This is achieved mainly through regression optimization on the dataset. Within a single episode, any changes to semantic adverbs require corresponding adjustments to the VLA model. After the VLA model predicts a suitable force, real-time reactive optimization is performed using the error between the measured force and the predicted force.
>
> Q5: Table 2 shows that binary grippers cannot achieve force control based on linguistic adverbs.  Dataset A serves as the standard group, Dataset B as the sliding scenario group, and Dataset C as the binary gripper control group. We strictly follow the LIBERO benchmark to first collect data on specific tasks, and then evaluate the model on these tasks.
>
> Q9: In realistic settings, the mapping from penetration depth to contact force is highly confounded by object stiffness, friction, local geometry, and contact location, so the same penetration depth does not imply the same force profile across objects and scenes. The tactile marker field directly provides a spatially distributed measurement of normal and shear deformations at the contact, which we found crucial for robust behavior across different objects and materials.

---

> > ### Author Rebuttal · Reviewer_tHpJ · 2026-03-31
> >
> > Thanks for the responses and clarifications. While the authors addressed several points in the rebuttal, the gap between the proposed tactile simulation and physical reality remains a primary concern. Specifically, the submission lacks empirical validation demonstrating that the advantages observed in the simulation framework translate to performance gains or increased fidelity in a real-world application.

---

> > > ### Author Response · Authors · 2026-04-04
> > >
> > > We are glad to hear that our response has helped address some of your concerns. Regarding your further comment, we believe there is still one important point that needs to be clarified.
> > >
> > > We would like to emphasize that **we did not propose any novel methods for tactile simulation or tactile sim2real** in this work. We directly adopted the existing setups from TacEX, FOTS, and Taxim.
> > >
> > > **We have supplemented the materials and experiments related to tactile sim2real. Link: (https://anonymous-tabero.netlify.app/static/pdfs/Tacsim2real.pdf)**. It includes the sim2real data of TacEX, FOTS and Taxim that we adopted, as well as the detailed implementation process of these methods in our work. We hope this can provide sufficient references for you and other reviewers to verify the reliability of our approach.
> > >
> > > In addition, we have also updated the sim2real results tested by ourselves, with full views presented in the anonymous webpage. At the bottom of the page shown in the anonymous link (https://anonymous-tabero.netlify.app/).
> > >
> > > Thank you again for your careful review of our paper and rebuttal.

---

### Official Review · Reviewer_kUiG · 2026-02-28

**Soundness:** 2
**Presentation:** 2
**Significance:** 3
**Originality:** 2
**Overall Recommendation:** 5
**Confidence:** 4

**Summary:**

This paper introduces Tabero, a benchmark and model suite for language-conditioned gentle robotic manipulation that explicitly incorporates closed-loop force feedback alongside vision and touch. The authors address the lack of scalable tactile datasets and evaluation protocols by proposing a data-efficient pipeline that repurposes open-source manipulation trajectories to generate diverse vision–tactile–language tasks, together with an evaluation metric that jointly measures task success and physical interaction quality. Building on this benchmark, they propose Tabero-VTLA, a Vision–Tactile–Language–Action architecture with a decoupled force–position interface executed by a hybrid controller, enabling real-time force-aware control. Experiments show that the proposed approach preserves task success while substantially reducing interaction forces under gentle manipulation instructions, demonstrating improved modulation of contact behaviors through multimodal feedback.

**Compliance With Llm Reviewing Policy:**

Affirmed.

**Final Justification:**

The clarification on tactile modalities and the Isaac Lab domain randomization setup addresses my concerns, I raise my score accordingly.

**Key Questions For Authors:**

- [IMPORTANT, score-relevant] Since the primary contribution lies on the proposal of benchmark. Please further demonstrate the reproducibility by uploading relevant code or provide other evidences for potential open-source possibility.
- Please further explain the reason why the performance of model with all modules is not shown in Table 3.
- Does proposed simulation supports adjust the physical parameters for objects (e.g, friction coefficient), which means a lot for domain randomization?

**Limitations:**

Please further discuss the potential negative societal impacts like misuse instead of just glossing over in the Impact Statement.

**Strengths And Weaknesses:**

Soundness:
- The claims are well-supported by further experiments empirically.

Presentation:
- The paper is clearly written and easy to follow.

Significance & Originality:
- This paper do propose a valuable benchmark and dataset for development intelligent robot with tactile.

---

> ### Author Rebuttal · Authors · 2026-03-31
>
> We appreciate your recognition of our work, and we promise to open-source our main projects in the future.
>
> Q1. Please refer to the following link for our code and supplementary evidence: https://anonymous-tabero.netlify.app/. We have provided an early version of our code in this link to demonstrate the basic task simulation, and the complete code will be open-sourced in the future.
>
> Q2. We regard 6D force, marker motion field, and tactile image as three independent tactile modalities. We compared the effects of different modalities in Table III. If all three modalities are to be input simultaneously, we believe an extra modality fusion mechanism is required to realize multi-modal tactile integration, which could be investigated as an independent research topic.
>
> Q3. Our simulation is built on NVIDIA Isaac Lab, which natively supports rich domain randomization over physical and visual parameters (e.g., friction, mass, restitution, lighting, textures). This allows us to collect diverse simulated data and expose the policy to a wide range of contact and dynamics conditions, which is crucial for improving sim‑to‑real robustness.
>
> In the present work, we primarily use this capability to randomize key physics parameters relevant to gentle manipulation, but the framework itself is general and extensible. In future work, we plan to leverage the same simulator for RL post‑training, using domain‑randomized rollouts to train a more generalized policy that can better adapt to novel objects and real‑world variations.

---

> > ### Author Rebuttal · Reviewer_kUiG · 2026-04-01
> >
> > Thank you for the rebuttal and the anonymous code page. The clarification on tactile modalities and the Isaac Lab domain randomization setup addresses my concerns. I have no further major questions and adjust my score accordingly.

---

> > > ### Author Response · Authors · 2026-04-08
> > >
> > > We sincerely appreciate your support and recognition for Tabero. Should you encounter any issues in our future open-source version, we remain committed to helping resolve them. We warmly welcome you to use our work.
> > >
> > > Once again, we sincerely thank you.
> > >
> > > With warmest regards,
> > > Authors

---

### Official Review · Reviewer_SRQm · 2026-03-12

**Soundness:** 4
**Presentation:** 3
**Significance:** 3
**Originality:** 3
**Overall Recommendation:** 4
**Confidence:** 4

**Summary:**

This paper builds a comprehensive simulation pipeline to generate tactile images and marker force fields for open-source robot trajectories. It introduces a VTLA model, which processes language commands that include specific requests about the applied force ("gently" or "firmly"). Additionally, the authors introduce new evaluation metrics that look beyond simple task success rates, incorporating average and maximum forces (MG, AG, MA, AA) to quantitatively assess the safety of the robot-object interaction.

**Compliance With Llm Reviewing Policy:**

Affirmed.

**Key Questions For Authors:**

Questions in “weakness” part above

6. Some parameters, such as the feedforward gain K_ff, can be difficult to tune. If the value is too large, it produces excessive force; if it is too small, the measured force (red line) constantly lags behind the predicted force. How can the tuning process for K_ff and other control parameters be standardized for different robot arms?

7. There is a lack of information about the exact number of trajectories or the total amount of demonstration data (in hours or episodes) used for VLA pre-training and fine-tuning.

**Limitations:**

Yes

**Strengths And Weaknesses:**

Strengths:
1. The introduction of linguistic commands to explicitly adjust the applied force in VLA-based manipulation is a novel and practical contribution.
2. Technical Soundness: The work demonstrates strong technical depth. It presents an end-to-end framework that covers scalable data generation, VTLA model training, low-level robotic arm and gripper control, and detailed ablation studies


Weaknesses also as questions:
1. Unfair Baseline Comparisons: Comparing the proposed method against baselines that only use raw 6D force vectors (e.g., Force E, Force D) may be an unfair comparison . The marker motion field contains much richer spatial information than a single 6-dof force vector. The paper lacks a comparison against VLA baselines that utilize this exact same spatial force field as their input.

2. While the paper integrates some baseline methods (Force E, IMG, Force D) into a unified framework for comparison, these ablation experiments are evaluated exclusively on the LIBERO-Object suite . They are not tested on the LIBERO-Spatial, Goal, or 10 suites.

3. Although the proposed VTLA achieves comparable results on LIBERO-Object, the baseline data shows that success rates on LIBERO-Spatial, Goal, and 10 drop significantly (often falling to 0.6 or below when using the tactile gripper) . What can be the reason that the performance on these specific task suites lags so kind of behind the original MuJoCo/Isaac baselines? and how to improve it?

4. Underutilization of Spatial and Shear Forces: The framework focuses on applying "gentle" or "firm" forces, but the low-level controllers (both the gripper force controller and the arm admittance controller) and the evaluation metrics (MG, AG, MA, AA) are based entirely on normal force. Shear force is ignored at the control level. Consequently, the rich spatial force field is underutilized; for instance, the shear force data could have been actively used to perform slip detection during the ultra-gentle tasks. How to better utilize spatial and shear force?


5. Lack of Real-World Validation: There are no real-world experiments to validate the system. It remains unclear whether the force prediction and control policies successfully transfer to physical tactile sensors and actual robotic hardware. For example, how to deal with the gap between simulated tactile and real tactile readings?

---

> ### Author Rebuttal · Authors · 2026-03-31
>
> We greatly appreciate your positive evaluation of our work, and your recognition serves as a strong motivation for us. Below are our responses to your questions.
>
> Q1. To the best of our knowledge, we have not found any VLA baselines that take the marker motion field as input other than our own baseline. Could you please provide some relevant examples? We will supplement the comparative results as much as possible.
>
> Q2 & Q3: During our experimental tests, we found that the failures in Libero Spatial, Goal, and 10 are highly related to the mechanical configuration of the gripper. We observed that when grasping a bowl, the collision body of the bowl is excessively thin, and the protective limit of the gripper prevents it from fully closing (to avoid collision between tactile sensors), making it difficult to grasp the bowl securely. Consequently, Libero Spatial has the lowest success rate, and similar issues exist in other task suites. To ensure fair evaluation of our model without being affected by physical structural constraints, we present the results on Libero Object, which has the best physical performance. Our recent tests show that these problems can be solved by replacing grippers with different physical configurations. The replacement methods and documentation will be provided for users in the open-source version of Tabero.
>
> Q4. In Dataset B with a gentle force setting of 10%, slippage occurs extensively across samples. Therefore, we conclude that the model can learn to leverage rich force field information for grasping under slipping conditions to a certain degree. At present, we are also considering adopting reinforcement learning to handle more extreme scenarios, and promoting the establishment of evaluation metrics for this research direction.
>
> Q6. The parameter  $K_{ff}$ indeed requires manual tuning. We consider this a common issue in the cross-embodiment field, and future research will focus on automating the tuning of these parameters using reinforcement learning.
>
> Q7.  We adopted the pre-trained weights from PI0, and conducted fine-tuning on our Tabero dataset.  Dataset A: 762 episodes; Dataset B: 726 episodes; Dataset C: 750 episodes.

---

> > ### Author Rebuttal · Reviewer_SRQm · 2026-04-03
> >
> > Thanks for the rebuttal and provded technical details, the some concerns are solved and concern like real world robot experiments still remain unresolved.
> >
> > Regarding to the answer to Q1: VLA like OmniVTLA[1] use image of tactile sensor, and Tactile-VLA [2] use normal force array as tactile information, although they are not marker motion field but also contain spaitial force information of gripper force, and can be more fair baselines that just use wrist 6D force vectors.
> >
> > [1] OmniVTLA: Vision-Tactile-Language-Action Model with Semantic-Aligned Tactile Sensing
> >
> > [2] Tactile-VLA: unlocking vision-language-action model's physical knowledge for tactile generalization

---

> > > ### Author Response · Authors · 2026-04-04
> > >
> > > Thank you for your constructive feedback on our work. We apologize for inadvertently omitting the real-world experiment results corresponding to Q5 in our rebuttal.
> > >
> > > Our real-world experiment results are provided via the anonymous **link**: https://anonymous-tabero.netlify.app/ These include tactile sim2real outcomes, comparisons between real-world and simulation scenarios, and additional analyses, all of which serve to demonstrate the transferability of our simulation framework.
> > >
> > > Regarding Q1: our comparative method Force D+FS corresponds to the reproduced results of Tactile-VLA. OmniVTLA employs tactile images, and our baseline IMG can reasonably reflect the performance of such vision-based approaches to a certain extent.
> > >
> > > We sincerely appreciate your recognition and constructive comments.

---

### Official Review · Reviewer_rwtw · 2026-03-13

**Soundness:** 3
**Presentation:** 2
**Significance:** 3
**Originality:** 3
**Overall Recommendation:** 5
**Confidence:** 4

**Summary:**

They introduce Tabero, a benchmark and model suite for gentle, language-conditioned robotic manipulation that demands fine-grained contact force perception. First, the Tabero benchmark addresses the scarcity of tactile data by presenting a data-efficient pipeline that repurposes open-source robot manipulation trajectories to generate a diverse set of vision-tactile-language tasks and establishes a multidimensional evaluation protocol that measures task success alongside physical interaction quality. Second, they propose Tabero-VTLA, a Vision-Tactile-Language-Action architecture featuring a decoupled force-position command interface; the resulting force-position commands are executed by a fixed hybrid controller to enable real-time, force-aware manipulation.

**Compliance With Llm Reviewing Policy:**

Affirmed.

**Final Justification:**

All primary issues addressed, and I updated my score to accept

**Key Questions For Authors:**

Figure4  is mentioned but the qualitative comparison details could be expanded to show failure cases, particularly where the method might over-smooth force.

**Strengths And Weaknesses:**

Strengths
The paper is written clearly, with well-defined technical details. The validation supports the primary claims, and the limitations are well discussed. The architectural contributions are effective, and the system demonstrates commendable empirical performance and robustness. The paper provides a comprehensive experimental evaluation across different scenarios. The proposed method appears technically sound, and the technical equation is generally correct

Weaknesses
Important implementation details missing from the manuscript. While extensive experiments are provided, the paper would benefit from a brief discussion comparing computational efficiency (inference time, FLOPs) with other methods to contextualize the latency limitation. Figure4  is mentioned but the qualitative comparison details could be expanded to show failure cases, particularly where the method might over-smooth force.   However, validation has several limitations:

The ablation research is limited and cannot fully analyze the contribution of individual components.
Failure cases were not discussed.
The robustness under different noise conditions has not been thoroughly evaluated.
Sensitivity to parameters has not been analyzed.

---

> ### Author Rebuttal · Authors · 2026-03-31
>
> We appreciate your recognition of our work. However, we would like to clarify your major concerns, especially regarding the ablation studies and failure cases.
>
> ### 1. Implementation Details
>
> We have provided extensive implementation details in the appendix. Additionally, we have supplemented our anonymous code and data assets at the following anonymous link: https://anonymous-tabero.netlify.app/.
>
> ### 2. Ablation Studies
>
> The custom-designed components of our work—including the integration of three tactile modalities and the closed-loop feedback control module—have been fully validated via ablation experiments in Table 3 and Figure 7. Furthermore, we performed complete ablation and analysis on our proposed benchmark dataset by splitting it into Groups A, B, and C (Table 2). All components proposed in this work have been examined with ablation studies. As for the VLA model architecture and tactile simulation, our implementation is directly built on Pi0, FOTS, and TacEX; the ablation analyses for these methods are thoroughly documented in their respective references.
>
> ### 3. Failure Cases
>
> For the dataset, we explain the causes of failed data collection in the second paragraph of Section 4.2 of the main paper, and Tables 12, 13, and 14 in Appendix C present the full dataset listings that include both successful and failed trials. For the model, we provided video demonstrations of failure cases in the supplementary video attachment of the original submission (see model_inference/task1_gentle_failure.mp4), and Figure 7(d) contains force-characteristic analyses of failure scenarios. We believe we have sufficiently presented failure cases from both qualitative and quantitative perspectives.
>
> ### 4. Robustness and Parameter Sensitivity
>
> Table 4 mainly presents experiments on the generalization performance to linguistic adverbs. The core VLA model architecture and tactile simulation parameters are directly adopted from our reference works (Pi0, TacEX, and FOTS). We consider that these studies have already conducted rigorous parameter-sensitivity analyses, so we directly used the parameters reported in those papers. For the inference efficiency of different methods, we show detailed model hyperparameters for all approaches in Tables 5 and 6 of Appendix A.
>
> The measured inference time on an NVIDIA RTX 4090 GPU is summarized below:
>
> | model part | inference time |
> | --- | --- |
> | image encoders | 14 ms |
> | observation forward pass | 32 ms |
> | x10 action forward pass (flow) | 27 ms |
> | network latency (off-board) | 13 ms |
> | total on-board inference | 73 ms |
> | total off-board inference | 86 ms |

---

> > ### Author Rebuttal · Reviewer_rwtw · 2026-04-03
> >
> > All primary issue’s addressed

---

> > > ### Author Response · Authors · 2026-04-08
> > >
> > > We sincerely appreciate your support and recognition for Tabero. We will provide comprehensive support in our future open-source versions, and we warmly welcome you to use our work.
> > >
> > > Once again, we sincerely appreciate your recognition.
> > >
> > > With warmest regards,
> > >
> > > Authors

---

### Decision · Program_Chairs · 2026-04-30

**Decision:**

Accept (regular)

**Comment:**

The reviewers broadly agree that the paper makes a valuable contribution by addressing the scarcity of vision-tactile-language data and proposing a sound framework for force-aware, language-conditioned robotic manipulation. The one dissenting reviewer (tHpJ) raised concerns about the sim-to-real gap and the lack of real-world experimental validation. In the rebuttal, the authors provided supplementary real-world demonstration data and clarified that the tactile simulation components are adopted from established prior works with documented sim-to-real performance. The AC finds that the authors' clarifications, combined with the supplementary evidence provided, are sufficient to address the core claims of the paper, and that the remaining concern falls partially outside the primary scope of the contribution.